# Microbiota and Serum Metabolic Profile Changes in Korean Native Hanwoo Steer in Response to Diet Feeding Systems

**DOI:** 10.3390/ijms232012391

**Published:** 2022-10-16

**Authors:** Jeong Sung Jung, Ilavenil Soundharrajan, Dahye Kim, Myunggi Baik, Seungmin Ha, Ki Choon Choi

**Affiliations:** 1Grassland and Forage Division, National Institute of Animal Science, Rural Development Administration, Cheonan 31000, Korea; 2Animal Genomics and Bioinformatics Division, National Institute of Animal Science, Wanju 55365, Korea; 3Department of Agricultural Biotechnology and Research Institute of Agriculture and Life Sciences, College of Agriculture and Life Sciences, Seoul National University, Seoul 08826, Korea; 4Dairy Science Division, National Institute of Animal Science, Rural Development Administration, Cheonan 31000, Korea

**Keywords:** grazing, housing, NGS, serum metabolic parameters, animal welfare

## Abstract

The diversity of bacteria and their function in cattle gastrointestinal tracts can influence animal welfare. Next-generation sequencing (NGS) was used to investigate microbial diversity in the feces of Hanwoo steers reared under natural grazing (GS) and housing (HS) systems. Additionally, serum metabolic parameters, such as liver and kidney markers and mineral and lipid content changes, as well as their correlation with pyrotags, were studied. A total of 6468 ± 87.86 operational taxonomic units (OTUs) were identified in both steer groups, of which 3538 ± 38.17 OTUs were from grazing steer and 2930 ± 94.06 OTUs were from GS. Chao1 index analysis revealed a higher bacterial richness in GS. The dominant bacterial taxa were *Bacteroidetes* and *Firmicutes*. GS showed lower *Bacteroidetes* and higher *Firmicutes* abundance than HS. The serum of HS showed consistent increases in gamma-glutamyl transpeptidase (γGTP), glucose (GLU), total cholesterol (T-CHO), and triglyceride (TG) levels. The impact of GS on animal health and serum metabolic markers was strongly correlated with microbiota. As shown in this study, grazing has a significant impact on the fecal microbiota at the phylum and family levels, as well as the serum biochemical metabolites of Hanwoo steers.

## 1. Introduction

The microbiota associated with the gut is important in maintaining the health of both animals and humans due to their symbiotic relationship. Several genes within the intestinal microbiota are involved in host health and survival, allowing for optimal energy from diets to be harvested and stored [1]. Genes also facilitate the production of vitamins, cofactors, or biologically active molecules associated with microbial activity, including short-chain fatty acids, indole, tryptamine, peptidoglycans, and lipopolysaccharides, which are essential to human health [2,3,4].

A major component of rural landscapes is livestock forming, particularly ruminants (sheep and cattle), which provide essential services to society, including improved soil health, biodiversity management, recreational activities, and community support [5,6]. As one of the most competitive and suitable feeding systems for cows globally, pasture grazing, in situ, has a low environmental footprint, is beneficial for animal health, and is relatively inexpensive to produce as well as to utilize [7]. In terms of animal welfare, it is believed that restricting animals’ access to resources may have an adverse effect on their health. It has been shown that animals with an inability to access natural feeding behaviors are more likely to develop stereotypical and other abnormal behaviors [8]. In a small study in Kerala, India, cattle restricted to accessing forages or grazing developed tongue rolling stereotypy, whereas cattle that were given free access to forages or grazing did not exhibit this condition [9]. Several studies have shown that grazing in diverse vegetation improves mineral intake balance, reduces oxidative stress, and improves antioxidant levels in cattle [10,11].

The impact of grazing on animal welfare may vary depending on grazing conditions, including pasture status and environmental factors. Animal health, digestibility, growth rate, and meat quality differ significantly between grazing and feedlot systems. A grazing animal has lower fatty acid content in its meat and higher levels of vitamins than a feedlot animal [12]. A previous report claimed that housing animals produce more polyunsaturated fatty acids [7], particularly omega-3 polyunsaturated fatty acids and conjugated linoleic acid, which would enhance the nutritional value of the product, while grazing animals have higher levels of total protein and casein as well as fat-soluble vitamins (β-carotene and α-tocopherol) in pasture-based organic milk [13,14]. In recent years, people have been expected to consume high-energy, balanced meals with a low-fat content for their health preferences [15]. Several studies have been conducted on the influence of the bovine microbiota on the host’s energy and metabolism. Moreover, recent next-generation sequencing (NGS) studies have demonstrated that the enteric microbiota shifts in response to animal performance across multiple species. Particularly in cattle, diets/feed additives/feed methods contribute to changes in rumen and fecal microbial communities [16,17,18,19], as well as animal welfare. Nonetheless, potential changes in the microbial communities of grazing cattle, including orchard grass, perennial ryegrass, Kentucky bluegrass, white clover, and tall fescue have not been thoroughly studied. Here, we investigated changes in the composition of the fecal microbiota community and the dynamics between the grazing and housing feeding systems of Hanwoo steers. In order to accomplish this, NGS was used to analyze the fecal microbiota of steers by targeting the V4 region of the 16S rRNA gene. Fecal samples were collected from housing and grazing animals after seven months and correlated with animal welfare and serum biochemical parameters.

## 2. Results

### 2.1. Physiological and Metabolic Testing of Hanwoo Steer

Grazing steers had an average final bodyweight of 415 kg, while housing steers had an average bodyweight of 376 kg. In comparison with their initial weight, the bodyweight of grazing steers (GS) increased by 47%. However, housing steers (HS) gained 30.7% of their weight. The average daily gain (ADG), feed intake, feed efficiency, and feed conversion ratio remained unchanged (Appendix A). The GS consumed more crude protein and total digestibility nutrients (TDN) than the HS (*p* < 0.01). We also examined the impact of GS and HS feeding systems on serum profiles, which are indicators of animal health. The following parameters were analyzed: albumin (ALB), creatinine kinase (CK), creatinine (CRE), gamma-glutamyl transferase (γ-GTP), glucose (GLU), serum glutamate oxaloacetate transferase (SGOT), serum glutamate pyruvate transferase (SGPT), lactate dehydrogenase (LDH), blood urea nitrogen (BUN), total bilirubin (T-Bil), and total protein (T-Pro); minerals such as calcium, phosphate, and magnesium; and lipid profiles, including total cholesterol (T-CHO), triglycerides (TG), and non-esterified fatty acids (NEFA). A month after the experimental trial, serum metabolic parameters had not changed significantly (Table 1). However, certain parameters such as ALB (*p* < 0.03), Ca (*p*
*<* 0.01), γ-GTP (*p* < 0.01), GLU (*p* < 0.03), SGOT (*p* < 0.09), LDH (*p* < 0.09), Mg (*p* < 0.03), phosphate (*p* < 0.05), T-CHO (*p* < 0.04), and TG (*p* < 0.03) were altered between the GS and HS after 3 months of the experimental trial (Table 1). After 5 months, GLU, γ-GTP, LDH, T-CHO, TG, and NEFA levels were significantly increased in the serum of GS compared to HS (Table 1). The serum levels of GLU, γ-GTP, Mg, T-CHO, TG, and NEFA were significantly higher in HS at the end of the experiment (Table 1) than in GS. Compared with GS, γ-GTP, GLU, T-CHO, and TG increased consistently in the serum of HS after three, five, and seven months of experimental trials.

### 2.2. Fecal 16S rRNA Gene Sequencing Report

Using Illumina sequencing, a total of 12 fecal samples from grazing and housing animals produced 3,214,142 raw reads. The CD-HIT-OTU/rDnaTools program was used to remove ambiguous, low-quality, chimera, and other sequences. Finally, 717,056 high-quality sequences were obtained from all fecal samples; the mean number of sequences per sample was 44,816 ± 5131 (mean ± standard deviation; the range was 36,381–56,347). OTU determination resulted in 6468 ± 87.86 OTUs across all samples (ranges between 384–637 OTUs). GS had an average of 589.66 ± 38.17 OTUs per sample, whereas HS had an average of 488.33 ± 94.06 OTUs per sample (Figure 1a). A Phred score above 20 (Q20%) was 98.0% and a score above 30 (Q30%) was 93.0%, which indicated that the used samples had a significant quality level that is essential for NGS analysis.

### 2.3. Sequencing Depth, Coverage, and Alpha Diversity Metrics

Appendix A presents a rarefaction analysis of the fecal microbiota of GS and HS. Based on the graph, the analyzed samples displayed a flatter curve for OTUs than the right curve, indicating that a significant number of reads was used for this analysis. According to the calculated good average, the sampling depth captured most of the species diversity, with a mean coverage of 0.998 ± 0.0003 per sample. In terms of alpha diversity metrics, both GS and HS had almost the same number and evenness of species (Figure 1b). As a measure of richness, the GS fecal sample had a higher Cho1 value than the HS fecal sample (650.9 ± 32.1 vs. 537 ± 105.4, respectively) (Figure 1a).

### 2.4. PCA Analyses of Grazing- and Housing-Related Fecal Microbiota Shifts

A principal component analysis (PCA) was performed in order to investigate changes in the microbiota in the fecal samples from GS and HS. Two and three components contributed approximately 64.69% and 77.0% to the overall variance, with the first component exhibiting the largest contribution (Figure 1c,d). This finding indicates that GS and HS feeding systems had a significant influence on the feces of steers.

### 2.5. Overall Fecal Microbiota Compositions of the Hanwoo Steers

The predominant phyla in both the GS and HS fecal samples were *Firmicutes* and *Bacteroidetes*, with the combined sequences of these two phyla accounting for 91.8% of the entire microbial population in both GS and HS (ranges 34.5–40.8% and 49.9–57.2% respectively) (Figure 2). The remaining sequences were classified as *Spirochaetes*, *Verrucomicrobia*, and unclassified bacteria, which accounted for less than 2% of the total sequences. Phylum-level data showed a higher *Firmicute* presence and a lower *Bacteroides* presence in the feces of the GS compared with the feces of the HS (*Firmicutes* 40.8 ± 3.0 vs. 34.5 ± 2.4%, respectively; *p* < 0.05; *Bacteroidetes* 49.9 ± 4.3 vs. 57.2 ± 4.4%, respectively; *p* < 0.047).

There was a higher percentage of *Ruminococcaceae* and *Lachnospiraceae* bacteria in fecal samples from GS compared with HS (20.9 vs. 16.9% and 5.1 vs. 4.2%, respectively, out of all *Firmicutes* sequences). Moreover, 6.8% of sequences were unclassified at the family level in the *Firmicutes* phylum, while all other families accounted for less than 8.0% in GS and 6.9% in HS. Within the *Bacteroidetes* phylum, *Bacteroidaceae* and *Sphingobacteriaceae* were the most prevalent families, followed by *Rikenellaceae* and *Prevotellaceae*. GS had lower levels of *Sphingobacteriaceae* (13.1 vs. 18.4 *p* < 0.009) and *Bacteroidaceae* (12.9 vs. 17.9% *p* < 0.002) than HS (Table 2). At the genus level, lower amounts of *Parapedobacter* (*p* < 0.04), and *Bacteroides* (*p* < 0.01) and higher *Porphyromonas* (*p* < 0.013), *Prevotella* (*p* < 0.001), *Ethanoligenens* (*p* < 0.004), and *Papillibacter* (*p* < 0.002) were observed in GS than in HS (Table 3). At the species level, *Parapedobacter koreensis*, *Paludibacter propionicigenes*, *Paludibacter propionicigenes*, *Ethanoligenens harbinense*, *Alistipes finegoldii*, and *Papillibacter cinnamivorans* significantly varied among the experimental steers (Table 4).

Next, we sought to determine if specific microbiota were associated with the effects of GS and HS feeding systems on serum markers, such as ALB, CK, CRE, G-GTP, glucose GLU, SGOT, SGPT, LDH, BUN, total bilirubin T-Bil, and TP; minerals, including phosphate, and magnesium; and lipid profiles, including T-CHO, TG, and NEFA. Significant correlations were found between the microbiota at the phylum/genus levels and serum clinical profiles. *Firmicutes* were positively associated with blood urea nitrogen and negatively associated with T-CHO and TG, whereas *Bacteroidetes* were positively associated with serum CA, P, ALB, and NEFA (Figure 3).

*Papillibacter* and *Coprococcus* were negatively correlated with total protein, calcium, and albumin at the genus level. CRE, GLU, and TG were negatively associated with *Barnesiella* and *Prevotella*. There were positive correlations between *Rikenellaceae* and CK, BUN, and γ-GTP at *p* < 0.05. The *Dorea* genus exhibited negative correlations with T-CHO, CA, ALB, mg2+, NEFA, and phosphate at *p* < 0.001. SGPT was negatively correlated with *Blautia*, *Lachnospiraceae*, *Phascolarctobacterium*, and *Achnospiraceae* but positively correlated with *Flavobacteriaceae*. *Ruminococcaceae* and *Clostridiales* were positively associated with T-Bil, whereas *Paraprevotella* and *Bacteroidales* were negatively associated with T-Bil (Figure 4).

## 3. Discussion

In this study, we report changes in the fecal microbiota of Hanwoo steers as a result of grazing pastures, as well as its impact on serum metabolic profiles and animal performance. Final bodyweight (kg) and average daily growth rate (ADG) did not change significantly compared with housing steers. There were significant differences between grazing steers (GS) and housing steers (HS) in terms of feed intake, feed efficiency, and conversion ratio. We found that the total intake of crude protein and total digestibility nutrients were higher for the GS than for the HS. The amount of hardly fermentable dietary fiber (ADF and NDF) found in grazing forages was higher than in diets fed to HS. As far as changes in physiological parameters in the serums of both HS and GS are concerned, short-term feeding systems did not result in any changes in any physiological parameters. However, ALB (*p* < 0.03), Ca (*p* < 0.01), γ-GTP (*p* < 0.01), GLU (*p* < 0.03), SGOT (*p* < 0.09), LDH *(p* < 0.09), Mg (*p* < 0.03), phosphate (*p* < 0.05), T-CHO (*p* < 0.04), and TG (*p* < 0.03) were altered between the GS and HS after 3 months of experimental trials. The levels of GLU, G-GTP, LDH, T-CHO, TG, and NEFA in the serum of GS were significantly higher than those of HS in trial month five. By the end of the experimental months, serum GLU, G-GTP, mg, T-CHO, TG, and NEFA levels were higher in HS than in GS. In comparison with grazing steers, G-GTP, GLU, T-CHO, and TG were the most consistently increased parameters in the serum of HS after 3, 5, and 7 months of experimental trials. Ruminant concentrate diets are a major source of glucose, either through an increase in propionate production in the rumen (a gluconeogenic precursor) or an increase in intestinal glucose absorption [20]. Gluconeogenesis produces glucose, which is the primary source of energy for ruminants [21]. Excess glucose is converted into fatty acids that circulate throughout the body, particularly in the adipose tissue of the body [22]. Increased energy intake leads to the increased production of propionate in the rumen and gluconeogenesis in the liver. It may be the principal reason for the high level of serum glucose and lipid metabolites in the serum of HS. A lack of energy causes an increase in NEFAs in cattle [23] or pathological problems such as ketosis and fatty liver [24]. In a state of negative energy balance, NEFAs are produced by the lipolysis of triglyceride, which is stored in adipose tissue and transported to other organs and tissues [25]. The hormone cortisol is an indicator of stress that promotes lipolysis and stimulates the production of NEFAs in the blood [26]. The present study found that the level of NEFA was consistently elevated throughout experimental periods in the serum of HS, confirming that the housing of animals without natural feeding behaviors is closely associated with some abnormal behaviors [8]. Limiting cattle’s exploratory or forage activities could result in a significant reduction in animal welfare and could explain the increase in NEFA levels found in the serum of HS. The present study did not analyze the level of stress-related markers. Consequently, it is necessary to determine the level of stress-related markers and their impact on serum metabolites in GS and HS. Experimental steers were tested for SGOT, SGPT, and γ-GTP serum levels, which are produced in hepatocytes and released into the bloodstream if hepatocytes are damaged by high-energy feeding and mold toxins. Steers fed with concentrate in housing conditions may suffer a significant negative impact on their liver markers when compared with steers that graze.

Alpha diversity indices (OTUs, Cho1) indicated that the fecal microbial diversity of GS was higher than that of HS. There has been research indicating that fiber-based diets improve microbial diversity because the fermentation of fiber stimulates microbial proliferation better than that of starch-based diets [27,28]. The fiber-based diet contains several secondary metabolites that can act as prebiotics and contribute to the improvement of bacterial diversity [29,30]. During this study, animals grazed a variety of pastures containing large amounts of non-structural carbohydrates (NSC) and non-fiber carbohydrates (NFC) in tall fescue orchard grass, perennial ryegrass, Kentucky bluegrass, and white clover [16,31]. Microbial diversity depends on both NSCs and NFCs. Due to this, the bacterial communities of the steers grazing on natural pastures were highly diverse. Additionally, steer that grazed received higher levels of crude protein and TDN, which might contribute to microbial proliferation, and forage varieties and biomass may influence the diversity of microbial communities in cattle.

A significant difference was observed between the two steer groups regarding the relative abundance of microbes. The majority of pyrotags in the fecal samples of GS and HS belong to the *Bacteroidetes* (49.90 ± 4.31 vs. 57.23 ± 4.4%, respectively; *p* < 0.047) and *Firmicutes* (40.85 ± 2.99 vs. 34.5 ± 2.4% respectively; *p* < 0.09). These phyla have previously been demonstrated to constitute the major gut-associated phylotypes in a variety of different mammalian species [16,17,19], suggesting that *Firmicutes* and *Bacteroidetes* (more than 90% of all high-quality bacterial pyrotags) are critical to the microbial ecology of mammalian guts. Phyla such as *Spirochaetes* and *Verrucomicrobia* accounted for less than 2% of the total sequences, while unclassified bacteria accounted for less than 1%.

It is essential to evaluate the *Firmicutes* to *Bacteroidetes* ratio in order to determine whether gut microbes have an effect on host energy needs [32]. GS had higher *Firmicutes* content (*p* < 0.05) and lower *Bacteroidetes* content (*p* < 0.009) in fecal samples compared with HS. This is consistent with what has been previously reported for other animals, including Angus steer [17,33] and yaks [16]. As compared with what we reported in the current study regarding other animals, there is a large variation in the ratio of *Firmicutes* to *Bacteroidetes* in cattle reported in previous studies [16,17,33]. According to these previous studies, the dominance of *Firmicutes* or *Bacteroidetes* is due to variations in diets, breed types, climate, and forming practices across a wide geographical range [34]. Furthermore, forage varieties, forage quality, and forage locations in grazing pastures appear to be more favorable for the abundance of *Firmicutes* or *Bacteroidetes* pyrotypes.

*Ruminococcaceae*, *Lachnospiraceae*, *Bacteroidaceae*, *Sphingobacteriaceae*, *Rikenellaceae*, and *Prevotellaceae* dominated both steer groups, with *Ruminococcaceae* and *Lachnospiraceae* being among the most abundant Firmicute phyla in GS compared with housing steers in the fecal sample. This study has demonstrated that these bacteria play a critical role in the degradation of starch and fiber, as well as in improving fiber digestibility [35]. *Ruminococcaceae* have also been reported to degrade protein [36]. Adequate nutrients were available in the grazing pastures, which were conducive to the relative abundance of fiber-degrading bacteria. *Ruminococcaceae* and *Lachnospiraceae* are important factors in stimulating the growth of fibrolytic bacteria and are found in the rumens of Holstein cows [37,38]. We found that *Sphingobacteriaceae*, *Bacteroidaceae*, *Prevotellaceae*, and *Rikenellaceae* represented the most dominant families in *Bacteroidetes*. The *Prevotellaceae* bacteria are a dominant bacterial species in the saccharolytic group in the rumen. They have a low protein binding ability and are capable of digesting a wide range of carbohydrate substrates [39]. HS feces contained slightly higher amounts of *Prevotellaceae*, indicating that the high carbohydrate ability could be attributed to a higher organic matter content. Recent studies have reported that Christensenellaceae, *Ruminococcaceae*, and *Rikenellaceae* play a major role in forage degradation in the rumen due to their strong adhesion to forage grass after incubation [40,41]. In the current study, higher proportions of *Ruminococcaceae* and *Lachnospiraceae* in GS are expected to improve fiber degradation. Among dominant genera, *Prevotella* (*p* < 0.001), *Ethanoligenens* (*p* < 0.004), *Papillibacter* (*p* < 0.002), *Coprococcus* (*p* < 0.002), *Dorea* (*p* < 0.002), and *Blautia* (*p* < 0.002) were the most abundant genera in GS. *Parapedobacter* (*p* < 0.04), *Bacteroides* (*p* < 0.01), *Alistipes* (*p* < 0.09), and *Porphyromonas* (*p* < 0.01) were dominant genera in the HS group. *Paraprevotella* was first identified in human faces and was found to produce succinic acid and acetic acid as end products of glucose metabolism in cattle [42]; it has also been found in pig and human feces [43,44,45]. It can utilize xylan as a growth substrate. It has been reported that *Paraprevotella* in the rumen of cattle fed with cornstalk (CS) may have some beneficial effects on CS NDF degradation [46]. The *Bacteroides* genus is another well-known intestinal bacterium that can be both helpful and harmful [47] and participate in the natural transmission of antimicrobial resistance genes [48]. It has the capability of hydrolyzing complex organic compounds [49], decomposing hemicellulose and xylan [50,51], and converting long-chain fatty acids into short-chain fatty acids [52]. The *Prevotella* genus, which belongs to the Alloprevotella group, has been characterized by significant genetic divergence; functional versatility; and is involved in the degradation of dietary proteins [53], the metabolism of peptides [54], and the utilization of hemicellulose [55]. In the rumen, *Treponema* is a common bacterial group that is associated with the degradation of soluble fibers [56]. The nutrient composition of the pasture and concentrate may favor the growth of fibrolytic bacteria.

In a final experiment, we sought to determine if there was a specific microbiota associated with the effects of GS and HS feeding systems on serum-marker minerals and if there was a significant correlation between the microbiota at the phylum–genus level and the serum clinical profiles. *Firmicutes* has been positively correlated with blood urea nitrogen and negatively correlated with T-CHO and TG at the phylum level, while *Bacteroidetes* has been positively associated with serum calcium, phosphate, ALB, and NEFA at the phylum level, at *p* < 0.05. It was observed that the abundance of *Firmicutes* in the feces of grazing steers was higher, which explained the role *Firmicutes* might play in influencing the lipid profile of these animals and, in particular, the levels of T-CHO and TG.

In terms of genus-level correlations, *Papillibacter* and *Coprococcus* were negatively correlated with total protein, calcium, and albumin. CRE, GLU, and TG were negatively associated with *Barnesiella* and *Prevotella*. *Rikenellaceae* was positively correlated with CK at *p* < 0.001, as well as with BUN at *p* < 0.05, and it was negatively related to γ-GTP at *p* < 0.05. The genus *Dorea* was negatively correlated with T-CHO, Ca^2+^ ALB, and mg^2+^ at *p* < 0.05 and NEFA and phosphate at *p* < 0.001. SGPT was negatively correlated with *Blautia*, *Lachnospiraceae*, and *Phascolarctobacterium*, whereas *Flavobacteriaceae* was positively correlated with SGPT at *p* < 0.001. T-Bil was positively correlated with *Ruminococcaceae* and *Clostridiales*, whereas *Paraprevotella* and *Bacteroidales* were negatively correlated. Microbiota and serum clinical parameters demonstrated strong correlations, especially with regard to the impact of grazing on animal health and serum metabolic markers.

## 4. Material and Methods

### 4.1. Pasture and Grassland Management

At the Daum Hanwoo farm in Jeongeup, Jeollabuk-do, South Korea, tall fescue (7.5 kg/ha), orchard grass (17 kg/ha), perennial ryegrass (3 kg/ha), Kentucky bluegrass (3 kg/ha), and white clover (2 kg/ha) were sown. Chemical fertilizers containing 21% nitrogen, 17% phosphoric acid, and 17% potassium were applied to the grassland. During the early spring, 20 bags of fertilizer were applied per hectare to the grassland. Following the first grazing, 15 bags of fertilizer per hectare were applied, followed by 5 bags per hectare for each subsequent grazing period (2nd to 5th). Following the last grazing, 10 bags of fertilizer were applied per hectare. A total of seven paddocks were used for the experimental study.

### 4.2. Animals and Feeding Systems

A study was conducted at the Daum Hanwoo farm in Jeongeup, Jeollabuk-do, South Korea. It was conducted in accordance with the animal care and standard guidelines of the National Institute of Animal Science, South Korea (approval number NIAS-2020-443). We recruited a total of twenty-six Korean native-breed cattle called Hanwoo steers for this study. The steers were representative of both the feedlot feeding system (*n* = 6; average initial bodyweight, 260.8 kg) and the grazing system (an average initial bodyweight of 219.8 kg; randomly, 6 steers (*n* = 6) were selected for data analysis out of 20 steers). Classification was made based on the practice of housing feedlot feeding—HS (concentrate and rice straw)—and grazing feeding—GS (pastures containing *tall fescue*, *orchard grass*, *perennial ryegrass*, *Kentucky bluegrass*, and *white clover*). The composition of feed and the concentrations of nutrients are presented in Appendix A. The animals in housing feedlots were fed rice straw and concentrate between 9:00 a.m. and 16.00 p.m. Hanwoo steers were grazed using a rotational grazing system in which each pasture was divided into five 0.5 ha each. Depending on the season’s forage growth, rotational pastures were grazed with differing grazing periods. There was an average grazing period of 3 to 8 days, followed by a rest period of 21 days in a rotational pasture. The growth of forage was slower during hot summer periods. Thus, grazing was reduced, and resting periods were lengthened (3–4 days grazing with 30–40 day resting periods). This resulted in a significant increase in forage regrowth. Grazing began in late April and ended in mid-November. A Hanwoo steer grazed from 8 a.m. to 6 p.m. The steer grazed for 24 days in May, 25 days in June, 10 days in July, 25 days in August, 23 days in September, 27 days in October, and 7 days in November. When there was a drought or heavy rainfall, grazing cattle were fed hay harvested from pastures containing tall fescue, orchard grass, perennial ryegrass, Kentucky bluegrass, and white clover. We also determined the bodyweight and average daily weight gain, feed efficiency, feed conversion ratio, and feed intake.

### 4.3. Fecal Sample Collection and Nutrient Analysis of Forages

Following a 12 h fast, fecal samples were collected via the rectum using a disposable glove and transferred to sterile cryogenic tubes. For microbiome analysis, samples were frozen in liquid nitrogen and stored at −80 °C. Pasture samples were collected at different times (May to November) from different locations within the same paddock before and after grazing and were taken to the laboratory for forage intake and chemical analysis. The samples were dried at 60 °C until a constant weight was achieved. We ground dry samples through a 1 mm screen to determine crude protein, acid detergent fiber (ADF), neutral detergent fiber (NDF), total digestible nutrient (TDN), and in vitro dry matter digestibility (IVDMD). Nutrient compositions of pasture from grazed fields are presented in Appendix A.

### 4.4. Blood Sampling and Metabolic Profile Test

After fasting for 12 h, blood was collected via the jugular vein using a classic needle and syringe after the sampling site had been cleaned with 70% alcohol. A serum-separating tube (SST) was used to collect the blood, and it was then transferred to the National Institute of Animal Science, Cheonan, Korea. The blood sample was allowed to clot at room temperature without being disturbed. Afterward, the clot was centrifuged at 3000rpm for 10 min to remove it. A biochemistry automatic analyzer (Hitachi 7180, Hitachi Ltd., Tokyo, Japan) was used to analyze serum biochemistry and minerals. Glucose (GLU), non-esterified fatty acids (NEFA), triglyceride (TG), total cholesterol (TCHO), total protein (Tpro), albumin (ALB), total bilirubin (Tbil), blood urea nitrogen (BUN), creatinine (CRE), serum glutamic oxaloacetate transaminase (SGOT), serum glutamic pyruvic transaminase (SGPT), alkaline phosphatase (ALP), gamma-glutamyl transferase (γGTP), lactate dehydrogenase (LDH), calcium (ca), magnesium (mg), and inorganic phosphorus (P) were measured using a Hitachi 7180 after calibration and quality control assessments with commercial enzyme assay kits from Wako (Fujifilm Wako Pure Chemical Ltd., Osaka, Japan). Globulin was calculated by subtracting albumin from total protein. All biochemical analysis was completed in a single day.

### 4.5. Genomic DNA Extraction

It was estimated that approximately 10 g of each sample was added to 10 mL of 0.01% Tween 20 in PBS in a sterile Stomacher bag. A sonicator was used to sonicate the mixture for ten minutes. A pellet was collected via centrifugation at 9000× *g* at 4 °C for 10 min, and DNA was extracted using the DNeasyPowerSoil Kit (Qiagen, Hilden, Germany) in accordance with the manufacturer’s instructions. Quant-IT PicoGree kit (Invitrogen, Waltham, MA, USA) was used to measure genomic DNA.

### 4.6. Library Construction and Sequencing

To amplify the V3 and V4 regions, sequencing libraries were prepared according to the protocols of the Illumina 16S Metagenomic Sequencing Library. A total of two nanograms of gDNA was amplified using PCR with 5 x reaction buffer, 1 mM dNTP mix, 500 nM universal forward and reverse primers, and Herculase II Fusion DNA Polymerase (Agilent Technologies, Santa Clara, CA, USA). First, the PCR was run for 3 min at 95 °C for heat activation, followed by 25 cycles of 30 s at 95 °C, 30 s at 55 °C, and 30 s at 72 °C, followed by a final 5 min extension at 72 °C. For the first amplifications, the universal primer pair with Illumina adapter overhang sequences was as follows:

V3-F: 5′-TCGTCGGCAGCGTCAGATGTGTATAAGAGACAGCCTACGGGNGGCWGCAG-3; V4-R: 5′-TCTCGTGGGCTCGGAGATGTGTATAAGAGACAGGACTACHVGGGTATCTAATCC-3

AMPure beads were used to purify the first PCR products (Agencourt Bioscience, Beverly, MA, USA). The first product was then amplified with PCR for further library construction containing the index using the NexteraXT Indexed Primer. For the second PCR, the cycling conditions were the same as the first PCR, with the exception of using 10 cycles. In order to quantify the final products, AMPure beads were used (KAPA Library Quantification kits for Illumina Sequencing platforms). The purified products were quantified using real-time quantitative PCR following the qPCR Quantification Protocol Guide (Agilent Technologies, Waldbronn, Germany) and qualified using the TapeStation D1000 ScreenTape system (Agilent Technologies, Waldbronn, Germany). The paired-end (2 × 300 bp) sequence was determined with Macrogen using the MiSeqTM platform (Illumina, San Diego, CA, USA). The poor-quality sequences were removed using CD-HIT-OTU/rDnaTools. In order to calculate the bacterial diversity in different groups, alpha and beta diversity indices were calculated from the complete OTU (operational taxonomic unit) table (alpha_diversity.py; UCLSUT/RDP (16S) or UNITE (ITS); alpha_rarefaction.py; make_2d_plots.py; and make_otu_heatmap_html.py).

### 4.7. Statistical Analysis

Student’s *t*-test was used to compare the microbiota and serum metabolic changes between the fecal samples of grazing and housing steers using SPSS 16.0 software (SPSS Inc., Chicago, IL, USA). Pearson correlation coefficients were generated using R software (Microgen) in order to understand the relationships between the bacterial taxonomic profiles and serum clinical parameters.

## 5. Conclusions

Grazing steers on natural pastures increased the diversity of bacterial communities in fecal microbiota at the phylum, family, and genus levels. Furthermore, Firmicute levels were higher in the feces of grazing steers on natural pastures than in the feces of housing steers. There are two dominant bacterial families in *Firmicutes*: *Ruminococcaceae* and *Lachnospiraceae*. Meanwhile, *Sphingobacteriaceae*, *Bacteroidaceae*, *Prevotellaceae*, and *Rikenellaceae* dominated the *Bacteroidetes*, with higher numbers among the housing steers. The changes in microbiota may have an impact on serum metabolic profiles (G-GTP, GLU, T-CHO, and TG) and feeding behavior. The findings of this study contribute to the current understanding of the gut microbiota of Hanwoo steers and provide evidence for the possible effects of various forages on the rumen microbiota of natural feeding animals.

## Figures and Tables

**Figure 1 ijms-23-12391-f001:**
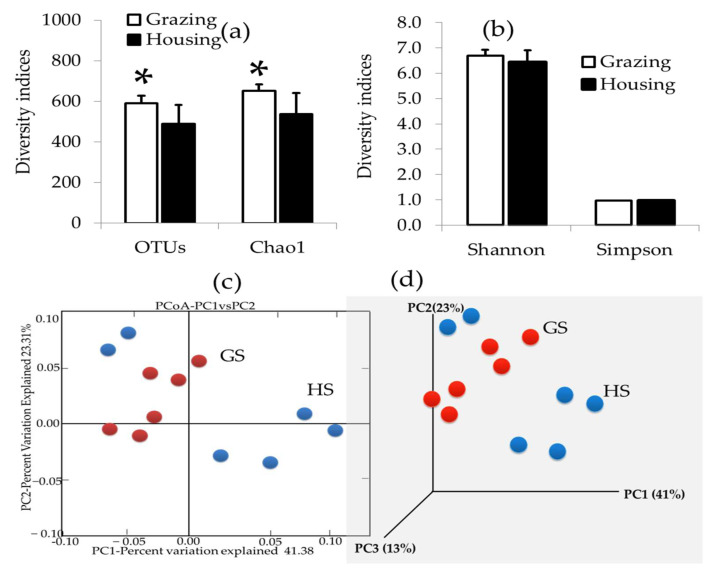
Diversity indices and principal coordinate plots, operational taxonomic units (OTUs), and level-weighted UniFrac distance between steer groups. (**a**) OTUs and Cho1 indices of bacterial diversity; (**b**) Shannon and Simpson indices of bacterial diversity; (**c**) 2D PCA of steer groups; (**d**) 3D PCA of steer groups. GS: grazing steer; HS: housing steer. * *p* < 0.05; grazing steer (GS) vs. housing steer (HS).

**Figure 2 ijms-23-12391-f002:**
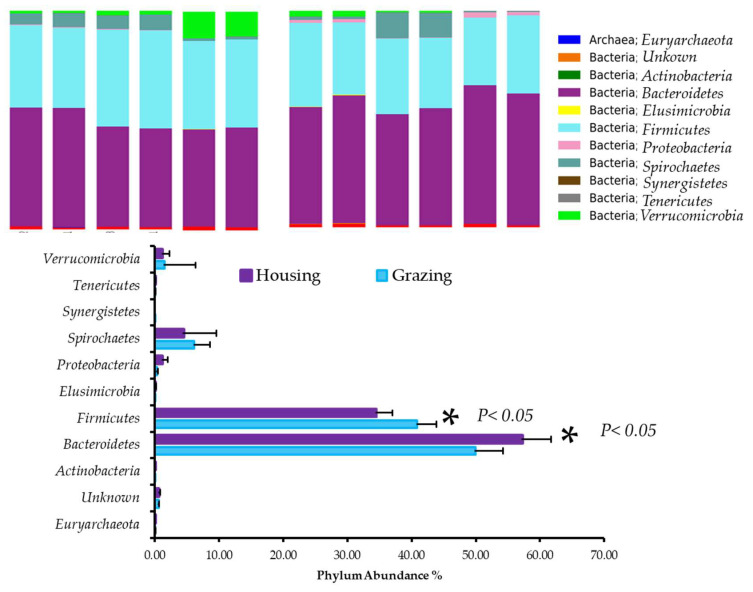
Relative abundance of microbiota changes in grazing and housing steers at the phylum level.

**Figure 3 ijms-23-12391-f003:**
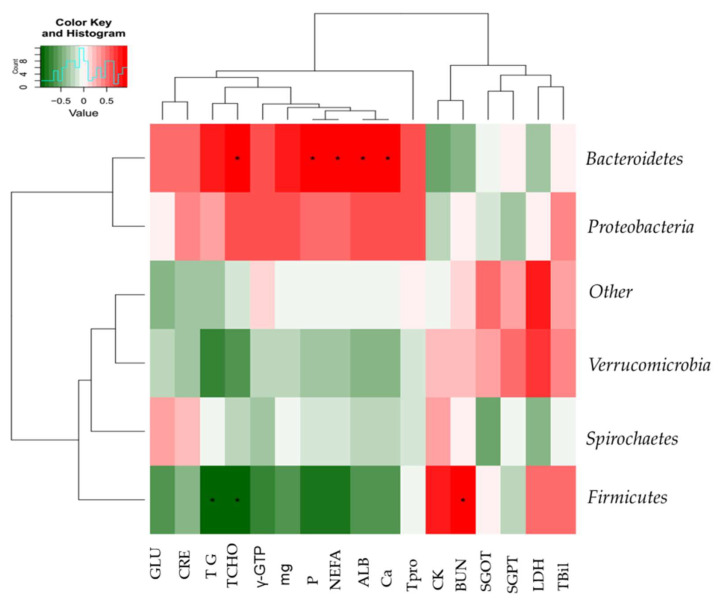
Heatmap correlation between the microbiota at the phylum level and serum metabolic profiles in experimental steers: Albumin (ALB), calcium (CA), creatinine kinase (CK), creatinine (CRE), gamma-glutamyl transferase (γ-GTP), glucose (GLU), serum glutamate oxaloacetate transferase (SGOT), serum glutamate pyruvate transferase (SGPT), lactate dehydrogenase (LDH), blood urea nitrogen (BUN), total bilirubin (T-Bil), total protein (TPro); minerals such as calcium (ca), phosphate (p), magnesium (mg); and lipid profiles including total cholesterol (T-CHO), triglycerides (TG), and non-esterified fatty acids (NEFA). * Indicates statistical significant difference between grazing and housing steers at 0.05 level.

**Figure 4 ijms-23-12391-f004:**
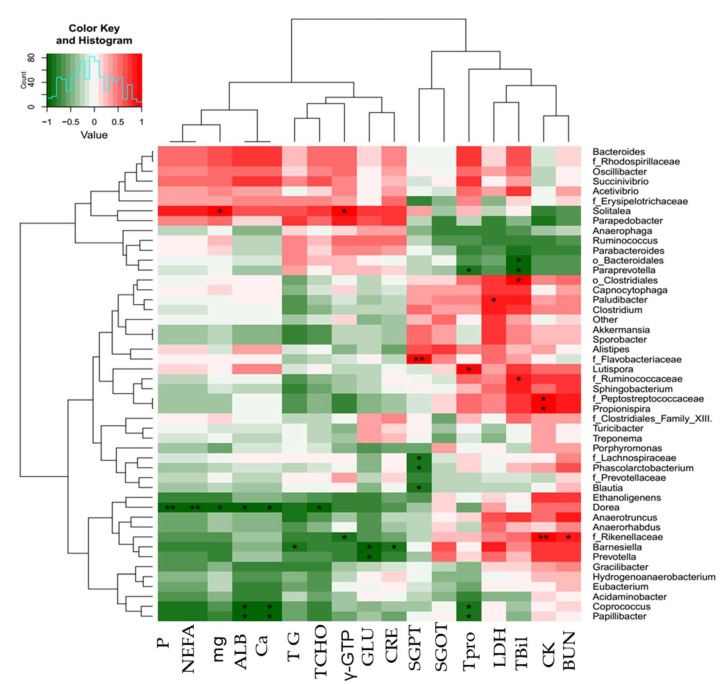
The correlation between genus-level microbiota and serum metabolic profiles in experimental steers represented by a heatmap: Albumin (ALB), calcium (CA), creatinine kinase (CK), creatinine (CRE), gamma-glutamyl transferase (γ-GTP), glucose (GLU), serum glutamate oxaloacetate transferase (SGOT), serum glutamate pyruvate transferase (SGPT), lactate dehydrogenase (LDH), blood urea nitrogen (BUN), total bilirubin (T-Bil), total protein (TPro); minerals such as calcium (ca), phosphate (p), magnesium (mg); and lipid profiles including total cholesterol (T-CHO), triglycerides (TG), and non-esterified fatty acids (NEFA)). * Indicates statistical significant difference between grazing and housing steers at 0.05 level. ** Indicates statistical significant difference between grazing and housing steers at 0.001 level.

**Table 1 ijms-23-12391-t001:** Physiological parameter changes in the serum of GS and HS in different periods.

Parameters	Groups	1M	3M	5M	7M
ALB (g/dL)	GS	5.10 ± 1.19	5.03 ± 0.82 *	5.93 ± 1.38	5.13 ± 0.33
	HS	5.23 ± 1.16	6.97 ± 0.17	6.57 ± 0.94	5.90 ± 0.54
CA (mg/dL)	GS	7.87 ± 2.12	7.57 ± 1.35 *	9.63 ± 2.52	7.20 ± 0.86
	HS	8.13 ± 2.37	11.7 ± 0.26	10.80 ± 1.55	9.43 ± 0.97
CK (mg/dL)	GS	138.0 ± 19.1	129.3 ± 27.8	174.6 ± 42.2	128.0 ± 14.9
	HS	189 3 ± 39.7	140.6 ± 20.9	182.3 ± 44.0	108.6 ± 4.50
CRE (mg/dL)	GS	0.03 ± 0.05	0.03 ± 0.05	0.00 ± 0.00	0.10 ± 0.08
	HS	0.03 ± 0.05	0.00 ± 0.00	0.07 ± 0.05	0.00 ± 0.00
γGTP (IU/L)	GS	20.33 ± 2.05	13.67 ± 0.47 *	24.0 ± 3.27 *	7.67 ± 3.86 *
	HS	27.33 ± 9.46	24.67 ± 1.70	32.33 ± 2.62	22.33 ± 2.62
GLU (mg/dL)	GS	100.6 ± 22.6	89.00 ± 10.2 *	96.67 ± 18.3 *	85.67 ± 5.56 *
	HS	95.67 ± 25.0	114.0 ± 3.74	111.6 ± 4.19	108.0 ± 3.56
SGOT (IU/L)	GS	70.67 ± 17.7	83.00 ± 1.41 *	101.0 ± 21.7	101.6 ± 27.0
	HS	103.3 ± 17.7	124. 6 ± 12.3	148.0 ± 46.1	83.33 ± 4.11
SGPT(IU/L)	GS	24.6 ± 5.46	23.6 ± 4.99	28.3 ± 7.72	26.33 ± 2.49
	HS	24.6 ± 5.44	31.0 ± 3.27	39.33 ± 2.05	30.67 ± 4.64
LDH(IU/L)	GS	1139 ± 232	1101 ± 243 *	1373 ± 180 *	1385 ± 76.11
	HS	1193 ± 368	1566 ±179	1831 ±70.29	1313 ± 103.1
Mg (mg/dL)	GS	25.3 ± 0.49	2.23 ± 0.41 *	25.3 ± 0.45	2.20 ± 0.16 *
	HS	2.47 ± 0.54	3.07 ± 0.12	3.10 ± 0.41	2.87 ± 0.25
*p* (mg/dL)	GS	11.0 ± 3.00	11.2 ± 1.02 *	12.3 ± 2.56	8.47 ± 0.62
	HS	11.2 ± 2.43	14.8 ± 1.25	13.17 ±1.26	11.90 ± 1.77
T-Bil (mg/dL)	GS	0.07 ± 0.05	0.07 ± 0.02	0.14 ± 0.07	0.22 ± 0.07
	HS	0.08 ± 0.04	0.08 ± 0.02	0.08 ± 0.06	0.14 ± 0.13
T-Cho(mg/dL)	GS	147.0 ± 52.4	119.3 ± 6.24 *	94.3 ± 10.8 *	130.0 ± 13.7 *
	HS	133.3 ± 29.3	164.3 ± 8.34	187.6 ± 29.2	188.6 ± 22.8
T-Pro(g/dL)	GS	1.53 ± 0.45	1.53 ± 0.26	1.67 ± 0.33	1.80 ± 0.08
	HS	1.43 ± 0.31	2.10 ± 0.00	2.00 ± 0.28	1.87 ± 0.17
TG (mg/dL)	GS	13.6 ± 5.91	12.3 ± 0.47 *	26.0 ± 2.16 *	16.33 ± 4.92 *
	HS	10.3 ± 1.89	22.3 ± 8.99	44.0 ± 2.16	37.33 ± 3.86
BUN (mg/dL)	GS	10.7 ± 2.13	12.3 ± 3.03	18.1 ± 3.25	15.60 ± 1.95
	HS	13.3 ± 4.01	14.7 ± 1.24	14.3 ± 2.73	12.03 ± 1.47
NEFA(μEq/L)	GS	132.6 ± 48.6	139.3 ± 47.2	201 ± 97.1 *	36.0 ± 19.1 *
	HS	229.0 ± 19.6	287.0 ± 28.0	257.3 ± 67.0	112.0 ± 23.2

M (month), albumin (ALB), calcium (CA), creatinine kinase (CK), creatinine (CRE), gamma-glutamyl transferase (γ-GTP), glucose (GLU), serum glutamate oxaloacetate transferase (SGOT), serum glutamate pyruvate transferase (SGPT), lactate dehydrogenase (LDH), blood urea nitrogen (BUN), total bilirubin (T-Bil), and total protein (TP); minerals such as calcium, phosphate, and magnesium; lipid profiles, including total cholesterol (T-CHO), triglycerides (TG), and non-esterified fatty acids (NEFA). * *p* < 0.05; grazing steers (GS) vs. housing steers (HS).

**Table 2 ijms-23-12391-t002:** Microbiota changes at the family level between grazing and housing steers.

S. No	Family	Grazing	Housing	STD	*p*-Values
1	*Sphingobacteriaceae*	13.1	18.4	2.56	0.009
2	*Flavobacteriaceae*	0.78	1.31	0.76	0.360
3	*Non-classified*	6.86	6.65	1.27	0.422
4	*Bacteroidaceae*	12.9	17.9	1.85	0.002
5	*Porphyromonadaceae*	2.25	1.57	0.87	0.250
6	*Prevotellaceae*	3.93	6.19	2.70	0.280
7	*Rikenellaceae*	4.60	6.20	1.47	0.140
8	*Streptococcaceae*	0.02	0.03	0.02	0.530
9	*Christensenellaceae*	0.12	0.11	0.05	0.940
10	*Clostridiaceae*	1.02	0.68	0.26	0.090
11	*Clostridiales* Family	1.11	0.89	0.39	0.480
12	*Clostridiales* Family III.	0.55	0.49	0.18	0.630
13	*Eubacteriaceae*	0.96	0.84	0.27	0.500
14	*Lachnospiraceae*	5.11	4.28	0.62	0.050
15	*Oscillospiraceae*	1.32	1.86	0.61	0.200
16	*Peptostreptococcaceae*	0.87	0.40	0.31	0.043
17	*Ruminococcaceae*	20.9	16.9	1.55	0.002
18	*Acidaminococcaceae*	0.33	0.20	0.17	0.274
19	*Selenomonadaceae*	0.14	0.09	0.06	0.330
20	*Kiloniellaceae*	0.00	0.04	0.02	0.050
21	*Rhodospirillaceae*	0.06	0.37	0.22	0.110
22	*Desulfovibrionaceae*	0.02	0.08	0.05	0.160
23	*Enterobacteriaceae*	0.04	0.13	0.10	0.310
24	*Succinivibrionaceae*	0.09	0.52	0.37	0.220

STD: Standard deviation.

**Table 3 ijms-23-12391-t003:** Modulation of Pyrotages at the genus level in grazing and housing steer feces.

Genus	Grazing	Housing	STD	*p*-Values
*Parapedobacter*	13.96	17.55	2.44	0.044
*Bacteroides*	12.76	16.84	2.03	0.014
*Porphyromonas*	3.173	1.572	0.81	0.013
*Paraprevotella*	3.068	5.296	2.54	0.250
*Prevotella*	0.694	0.083	0.17	0.001
*Alistipes*	4.237	6.059	1.44	0.090
*Ethanoligenens*	2.880	1.774	0.45	0.004
*Papillibacter*	9.173	6.020	1.23	0.002
*Coprococcus*	1.069	0.605	0.20	0.021
*Dorea*	0.281	0.145	0.07	0.02
*Blautia*	0.187	0.127	0.08	0.020
*Treponema*	4.467	4.471	3.75	0.990
Non-classified	4.113	4.286	0.51	0.610

**Table 4 ijms-23-12391-t004:** Species-level changes in the fecal microbiota of steers in response to diet systems.

S. No	Species Name	Grazing	Housing	STD	*p*-Values
1	Non-classified	1.264	1.153	0.003	0.510
2	*Parapedobacter koreensis*	4.351	6.196	0.006	0.002
3	*Parapedobacter soli*	12.14	11.35	0.030	0.700
4	*Muribaculum intestinale*	3.268	5.329	0.014	0.090
5	*Paludibacter propionicigenes*	1.793	0.263	0.002	0.000
6	*Bacteroides cellulosilyticus*	0.000	0.252	0.002	0.150
7	*Bacteroides clarus*	3.809	2.655	0.014	0.220
8	*Bacteroides plebeius*	4.665	5.339	0.018	0.600
9	*Porphyromonas pogonae*	2.062	1.406	0.009	0.270
10	*Paraprevotella clara*	3.068	5.296	0.025	0.250
11	*Prevotella shahii*	0.455	0.000	0.002	0.019
12	*Alistipes finegoldii*	1.110	2.329	0.009	0.069
13	*Alistipes onderdonkii*	1.161	1.138	0.005	0.940
14	*Alistipes putredinis*	0.993	1.283	0.005	0.440
15	*Flavonifractor plautii*	0.748	0.505	0.003	0.220
16	*Intestinimonas butyriciproducens*	2.519	2.409	0.006	0.810
17	*Acidaminobacter hydrogenoformans*	1.112	0.894	0.004	0.480
18	*Kineothrix alysoides*	2.128	1.746	0.006	0.420
19	*Eubacterium] tenue*	0.638	0.281	0.002	0.450
20	*Clostridium] cellobioparum*	1.298	0.766	0.005	0.130
21	*Clostridium] stercorarium*	0.141	0.340	0.001	0.070
22	*Ethanoligenens harbinense*	2.880	1.774	0.004	0.004
23	*Papillibacter cinnamivorans*	9.173	6.020	0.012	0.002
24	*Treponema porcinum*	3.584	3.973	0.038	0.880
25	*Akkermansia glycaniphila*	4.901	1.175	0.030	0.130

STD: Standard deviation.

## Data Availability

The experimental data are available upon request from the corresponding author.

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
