# Peer review of "Microbiota and Serum Metabolic Profile Changes in Korean Native Hanwoo Steer in Response to Diet Feeding Systems"

_ijms, 2022, doi:10.3390/ijms232012391_

Round 1

Reviewer 1 Report

This study said that the gut microbiota of Hanwoo steers and how different forages might affect the rumen microbiota of animals that eat in the wild. The microbiota was strongly linked to how the grazing system affected the health of the animals and the metabolic markers in their blood. It is hard to say the weakness: this is a smooth study and the objectives seem to be reasonable. Though the methods and contents are conventional and flat we can consider it reasonably new.

 Comments

1. Line 26: idex à index.

2. Line 63: what is “[7] reported”, please modify this sentence.

3. Line 67: The preferred citation format for [13] and [14] is [13, 14]..

4. Line 82: 16SrRNA gene à 16S rRNA gene; same in Line 124.

5. Line 114: The legend of Figure 1; one month trial, three months trail, five months trial; seven months trial; Please unify the feeding period units throughout the entire manuscript.

6. Line 126-128: This sentence should be clarified.

7. Line 131-132: make it clear “per sample & per animal.

8. Line 132: Figure 1a is inconsistent with this explanation.

9. Line 148: Please unite the decimal point in these numbers (64.69% and 77.0 %), and throughout the entire text and tables, all decimal point numbers should be unified..

10. Line 149: Figure 2c & d.

11. Line 153: Figure legend, please use the same capitalization in figure numbers.

12. Line 158: Grazing and Housing

13. Line 182: Ethanoligenens harbinense, Alistipes finegoldii

14. Table 1: Except for bacterial names, italics should not be used.

15. Table 2. Non-classified; not italic (same in Table 3).

16. Figure 4. Bacterial names in the figure better to present in italic (same in Figure 5).

17. I would strongly advise authors to revise their entire manuscripts again in great detail.

Author Response

We thank the reviewer for providing useful comments about our research paper, which will be used to improve the quality of the manuscript. We ask apology for typographical mistakes and grammatical errors.   In response to the reviewers' suggestions, we have read through the entire manuscript and modified it accordingly. Please note that changes have been made in red across the manuscript.

  1. Line 26: idex à index.

Thank you, the error has been corrected 

  1. Line 63: what is “[7] reported”, please modify this sentence. Line 67: The preferred citation format for [13] and [14] is [13, 14].

Thank you very much for your kind suggestion. The reviewer's recommended lines have been revised as follows:  Previsous report claimed that the housing animals produced more polyunsaturated fatty acids [7], particularly omega-3 polyunsaturated fatty acids, and conjugated linoleic acid, which would enhance the nutritional value of the product, while grazing animals had higher levels of total protein and casein  as well as fat soluble vitamins (β-carotene and α-tocopherol) in pasture based organic milk [13,14].

  1. Line 82: 16SrRNA gene à 16S rRNA gene; same in Line 124.

Throughout the manuscript, space has been provided between 16S and rRNA.

  1. Line 114: The legend of Figure 1; one month trial, three months trail, five months trial; seven months trial; Please unify the feeding period units throughout the entire manuscript.

The figure 1 has been modified into a table in accordance with the suggestion of another reviewer. All tables and figures now have clear legends.

  1. Line 126-128: This sentence should be clarified.

Thank you for sharing this valuable information. The following sentences harephrased as follows:ified:   Using Illumina sequencing, a total of 12 fecal samples from grazing and housing animals produced 3214142 raw reads. CD-HIT-OTU/rDnaTools program was used to remove ambiguous, low-quality, chimera, and other sequences. Finally, 717,056 high quality sequences were obtained from all fecal samples; the mean number of sequences per sample was 44816 ± 5131 (mean ± standard deviation; the range was 36381-56347). 

  1. . Line 131-132: make it clear “per sample & per animal.

Yes, it is OTUs per samples not per animals.  Thanks for kind information.

  1. Line 132: Figure 1a is inconsistent with this explanation (GS had an average of 66 ± 38.17 OTUs per sample whereas HS had an average of 488.33 ± 94.06 OTUs per sample (Figure 1a)

Yes, we have verified the data presentation. It contains typographical errors. The error has been corrected as follows:  OTUs determination resulted in 6468 ± 87.86 OTUs across all samples (ranges between 384 - 637 OTUs). GS had an average of 589.66 ± 38.17 OTUs per sample whereas HS had an average of 488.33 ± 94.06 OTUs per sample

  1. . Line 148: Please unite the decimal point in these numbers (64.69% and 77.0 %), and throughout the entire text and tables, all decimal point numbers should be unified.

In accordance with the reviewers' suggestions, we have revised and unified the data presentation throughout the manuscript.

  1. Line 149: Figure 2c & d. 11. Line 153: Figure legend; please use the same capitalization in figure numbers.

As per the reviewer's suggestions, unified figure numbers have been provided.

  1. Line 158: Grazing and Housing

It has revised as Relative abundance of microbiota changes in Grazing and Housing steers at phylum level

  1. . Line 182: Ethanoligenens harbinense, Alistipes finegoldii

It has revised as At species level, Parapedobacter koreensis, Paludibacter propionicigenes, Paludibacter propionicigenes, Ethanoligenens harbinense,  Alistipes finegoldii and Papillibacter cinnamivorans significantly varied among the experimental steers

  1. . Table 1: Except for bacterial names, italics should not be used. Table 2. Non-classified; not italic (same in Table 3).

We have agreed with the reviewer's comments and revised the manuscript accordingly.

  1. Figure 4. Bacterial names in the figure better to present in italic (same in Figure 5).

All bacterial names have been presented in italics throughout the manuscript 

  1. I would strongly advise authors to revise their entire manuscripts again in great detail.

I would like to thank you for your positive comments regarding the submitted article. The manuscript has been revised in accordance with the suggestions of the reviewer

Reviewer 2 Report

This is a study presenting novel information on fecal microbiota and serum markers in steer reared under two different feeding systems. Natural grazing (GS) showed a lower Bacteroidetes and a higher Firmicutes abundance than housing system (HS), using next-generation sequencing (NGS). Furthermore, GS increased the diversity of bacterial communities in fecal microbiota at the phylum, family, and genus levels.

Comments

1. Although the study is novel and well-conducted, the article is full of typos and inconsistencies which make it difficult to follow. I recommend that a native English speaker is hired to correct all the typos throughout the manuscript.

2. Abbreviations should follow a logic based on the most widely accepted use. For example, the authors use the abbreviation OUT to describe Operational Taxonomic Unit, when the generally accepted abbreviation is OTU. There needs to be a polishing in all minor details throughout the whole manuscript, even in the References. Abbreviations should be explained as footnotes in each Figure and Table.

3. The Abstract ends with the conclusion "A study presents novel information on fecal microbiota and serum markers in steer reared under two different feeding systems". This is not actually a Conclusion but the background. The authors need to rewrite their manuscript in a way that there is a flow in the reading.

4. Figure 1 makes no sense at all as there are different units in each variable and they are all presented as if the vertical axis refers to the same unit. I recommend that the Figure should be replaced by a Table to avoid such confusion and to clarify the difference.

5. The authors should report the exact P-values (and not for example P<0.01) in Tables so that the readers can understand the difference between the mean values in each group.

Author Response

We thank the reviewer for providing useful comments about our research paper, which will be used to improve the quality of the manuscript. We ask apology for typographical mistakes and grammatical errors.   In response to the reviewers' suggestions, we have read through the entire manuscript and modified it accordingly. Please note that changes have been made in red across the manuscript.

  1. Although the study is novel and well-conducted, the article is full of typos and inconsistencies which make it difficult to follow. I recommend that a native English speaker is hired to correct all the typos throughout the manuscript.

Thank you for your valuable and positive comments regarding our research article. With the assistance of native English speakers and an online tool, the language of the manuscript has been revised.

  1. Abbreviations should follow a logic based on the most widely accepted use. For example, the authors use the abbreviation OUT to describe Operational Taxonomic Unit, when the generally accepted abbreviation is OTU. There needs to be a polishing in all minor details throughout the whole manuscript, even in the References. Abbreviations should be explained as footnotes in each Figure and Table.

We have strongly agreed with the reviewer's comment and have provided it in all figures and tables. Albumin (ALB), calcium (CA), creatinine kinase (CK), creatinine (CRE), gamma-glutamyl- transferase (γ-GTP), glucose (GLU), serum glutamate oxaloacetate transferase (SGOT), serum glutamate pyruvate transferase (SGPT), lactate dehydrogenase (LDH),blood urea nitrogen (BUN), total bilirubin (T-Bil), total protein (TPro), minerals such as calcium (ca), phosphate (p), magnesium (mg) and lipid profiles includes total cholesterol (T-CHO), triglycerides (TG) and non-esterified fatty acids (NEFA).  

  1. The Abstract ends with the conclusion "A study presents novel information on fecal microbiota and serum markers in steer reared under two different feeding systems". This is not actually a Conclusion but the background. The authors need to rewrite their manuscript in a way that there is a flow in the reading.

Thank you for providing valuable information regarding an improvement to the submitted manuscript. In the abstract, the conclusion was revised as follows:  As shown in this study, grazing has a significant impact on the fecal microbiota at the phylum and family levels, as well as the serum biochemical metabolites of Hanwoo steers. In addition, the whole manuscript has been revised significantly and improved its data presentation and writing for easily understanding by the readers.

  1. Figure 1 makes no sense at all as there are different units in each variable and they are all presented as if the vertical axis refers to the same unit. I recommend that the Figure should be replaced by a Table to avoid such confusion and to clarify the difference.

In response to the reviewer's comment, we have replaced figure 1 with table 1. 

  1. The authors should report the exact P-values (and not for example P<0.01) in Tables so that the readers can understand the difference between the mean values in each group.

           Yes, we have provided exact P values for all experimental data based on SPSS analysis.

Round 2

Reviewer 2 Report

There are still several typos and inconsistencies.